# Reuse of KOH Solutions during Black Ripe Olive Processing, Effect on the Quality of the Final Product and Valorization of Wastewaters as Possible Fertilizer Product

**DOI:** 10.3390/foods11121749

**Published:** 2022-06-14

**Authors:** Pedro García-Serrano, Manuel Brenes, Concepción Romero, Pedro García-García

**Affiliations:** Instituto de la Grasa (CSIC), Campus Universidad Pablo de Olavide, 41013 Sevilla, Spain; pedgarser@gmail.com (P.G.-S.); brenes@ig.csic.es (M.B.); c.romero@csic.es (C.R.)

**Keywords:** black olive, KOH, reuse, valorization, wastewater

## Abstract

A high volume of water is needed to produce black ripe olives, which also entails a significant volume of wastewater with a high organic and inorganic contaminant charge. To reduce this problem, the reuse of KOH solutions (lyes) in a new process was studied. Once the lyes were removed from the tanks, KOH was then added for a new darkening process. Reusing the lyes up to four times gave rise to a product with similar physico–chemical and organoleptic characteristics as obtained with fresh solutions. The application of this process reduced coadjutant consumption by 32% and water by 20%, while global wastewater presented a high K content whose concentration could be valorized as a fertilizer by replacing commercial potassium nitrate.

## 1. Introduction

“Black ripe olives”, also known as California-style olives, is the name by which “black olives by oxidation” are known [1]. They are very attractive to consumers due to their shiny black color, and are used for salads, pizzas, sandwiches and many other fast food products.

The industrial processing of black ripe olives in Spain includes a storage stage in fiberglass containers (10,000 kg of olives and 5500 L of liquid) under acidic conditions for months [2], followed by a darkening step in horizontal cylindrical tanks in which the olives (10,000 kg) are treated with 10,000 L of several dilute solutions of sodium hydroxide (lye). After each alkaline treatment, olives are put in water and air is bubbled in during the whole process. During this time, the olives develop a black color, and they are then submerged in a ferrous solution (gluconate or lactate) to prevent color deterioration during packaging. Finally, the olives are canned in airtight containers and sterilized [3].

A high volume of water is needed to produce black ripe olives (4–7 L/kg), which also presupposes a high volume of wastewater [4]. To mitigate this problem, many Spanish manufacturers carry out a single alkaline treatment, followed by two washes and, finally, the black olives are placed in a ferrous solution to fix the color that has been formed [5].

There have been previous studies aimed at reusing the different liquids from the process. For example, it was proven that the preservation solution can be reused in a new process without affecting the quality of the final product [6], and that it can be employed after the main alkaline treatment alongside the first washing [7]. Although several Spanish companies have implemented these new methods, they are not widely used worldwide. Moreover, the reuse of the fixing color solution after purification has been studied at laboratory scale for a new process and as a packing cover brine [8].

Despite all these measures, wastewaters from black ripe olive processing still represent a big environmental problem for factories due to their high organic and mineral content in sodium that limits their use for agricultural purposes [9]. Nevertheless, it has recently been verified that the substitution of NaOH by KOH during the oxidation stage of olives allows the use of lye solution and washing waters as organic fertilizers without affecting the physico–chemical and sensory characteristics of the product [10,11].

Reuse of NaOH solutions is common practice in Spanish table olive factories but reuse of the new KOH solutions has never been studied, and is the aim of this study. It must be noted that the physico–chemical and organoleptic characteristics of the final product could be affected by the number of reuses. In addition, the concentrates obtained from the evaporation of the global KOH wastewaters are assessed, taking into consideration their further use as fertilizers.

## 2. Materials and Methods

### 2.1. Olive Processing

The experiments were carried out with olives of the Hojiblanca cultivar stored for eight months under acidic conditions [2]. The darkening process consisted of placing 3 kg of olives in each cylindrical container (20 cm Ø, 40 cm long) with 3 L of liquid (lye, washing water or ferrous gluconate solution). Air (0.2 m^3^/h) was bubbled along the horizontal bottom of the cylinder (20 diffusers).

First, the olives were immersed in 0.75 M KOH, and, once the alkali reached the pit, the solution was removed and stored. Subsequently, the fruits were placed into a solution of tap water/preservation liquid (1:1) for 20 h [7]. Then, the fruits were put in a new washing solution (tap water) for another 24 h aeration cycle. The pH of the liquid was controlled at 8.3 units by a pH meter (pH/mv Controller 252, Crison, Alella, Spain) connected to an automatic dosage system which bubbled CO_2_ during the washing stages in order to neutralize the residual alkali present in the olive flesh. The two washing waters were collected together with the lye. On the third day, the fruits were covered with a ferrous gluconate solution (1 g/L) and aerated for another 5 h to fix the black color formed [7]. The weight of the olives was also measured before the lye treatment and after applying the fixing solution.

Finally, 145 g of pitted olives and 175 mL of cover brine (0.15 g/L ferrous gluconate and 35 g/L NaCl) were placed in A-314 jars (Juvasa, Dos Hermanas, Spain). The jars were closed and sterilized at 121 °C in a computer-controlled Steriflow retort (Madinox, Barcelona, Spain) to reach a lethality value of 15 F_0_ [1]. The jars were kept at ambient temperature and the physico–chemical and sensory characteristics of the olives were measured after two months.

### 2.2. Experimental Design

The lyes and washing waters of two tanks (replicates) from one oxidation process (Control) were stored at 5–7 °C for further vacuum evaporation (Figure 1).

This process was repeated in two other two tanks, and the residual concentration of KOH in the lye was analyzed by titration with 0.2 M HCl to enrich it with fresh KOH, in order to reach the fixed 0.75 M KOH concentration. It was repeated three more times and the final KOH solution along with all the washing waters generated were gathered and stored at 5–7 °C for further vacuum evaporation.

Evaporation was carried out in a rotary evaporator (Büchi Rotavapor model RE-114, Flawil, Switzerland) under vacuum at 60 °C to 10% of its initial volume, and the pH was then dropped to about 5.5–6.0 units by adding HNO_3_ (65%, *w*/*w*) [12].

### 2.3. Physico–Chemical Analyses of Olives

The superficial color of olives was expressed as reflectance at 700 nm (R_700_) measured using a BYK-Gardner model 9000 Color-view spectrophotometer (Silver Spring, MD, USA). Lower reflectance values indicated darker fruit. The data from each measurement was the average of 10 olives.

Firmness was expressed as the force (N) required to break 100 g of pitted olives in a Kramer shear-compression cell coupled to a Texture Analyzer TA.TX plus (Stable Microsystems, Godalming, UK). The crosshead speed was 200 mm/min. Measurements were performed on three pitted olives and the data was the average of 10 determinations.

The sensory characteristics of olives were tested according to the “Method for sensory analysis of table olives” [13] in the normalized testing room of the Instituto de la Grasa. This method classifies olives commercially through the use of descriptors related to the perception of negative sensations (“abnormal flavor”), gustatory attributes (salty, bitter and acidic) and kinesthetic sensations (hardness, fibrousness and crunchiness). The statistic used to indicate the values of the attributes was the median of the individual data of the eight training testers and the variability by robust standard deviation.

Triangular comparisons were also made to detect differences in flavor between processed olives using fresh lye (initial) or after four reuses. Three samples (two the same and one different) were presented simultaneously to each taster who was asked to identify the single sample. Each panelist performed six tests with the samples presented in all possible positions [14]. Then, for each comparison (8 testers × 6 positions), 48 determinations were made.

### 2.4. Analysis of Minerals in Olives and Liquids

One mL of liquid or 1 g of olive paste was digested in DigiPREP equipment (SCP Science, QC, Canada) with 25 mL of 14 M HNO_3_ at 120 °C for 8 h, nitric was evaporated at 140 °C after the addition of 5 mL solution HClO_4_/HNO_3_ (4/1). Subsequently, the solution was put into a 25 mL graduated flask and filled with deionized water and diluted to obtain a concentration lower than 200 meq/L of sodium or potassium in order to be determined in a Metheor flame photometry (model NAK-1, PACISA, Madrid, Spain).

Nitrogen and carbon were analyzed by elemental analysis, using a LECO CHNS-932 analyzer (St Joseph, MI, USA), after drying the sample at 105 °C and the moisture calculated.

### 2.5. Chemical Analyses of Liquids and Concentrates

Total solids were determined according to 2540 methods standard procedures [15], the density analyzed at 20 °C with a 0.1 L volumetric flask, the viscosity analyzed with a viscometer Ostwald at 20 °C, the water activity analyzed using Aqualab equipment (Decagon Devices, Inc., Pullman, WA, USA), and the pH was measured using a Crison model 2001 pH meter (Crison Instruments, Barcelona, Spain).

Sugars were analyzed by HPLC with a Rezex RCM-Monosaccharide Ca^+^ (8%) column (300 × 7.8 mm i.d., Phenomenex, Torrance, CA, USA) held at 85 °C, deionized water as eluent at 0.6 mL/min, and the detection was performed with a Waters 410 refractive index detector [16]. Organic acids and ethanol were analyzed using the same detector with a Spherisorb ODS-2 (5 μm, 250 × 4.6 mm, Waters Inc., Mildford, MA, USA) column with deionized water (pH adjusted to 2.3 with phosphoric acid) as mobile phase [17]. The same column was used to separate the phenols that were identified by a Waters 996 diode array detector (Waters Inc., Mildford, MA, USA) using an elution gradient with water (adjusted to pH 3.0 with phosphoric acid) and methanol [18].

### 2.6. Statistical Analyses

Comparisons of the different mean values of the physico–chemical determinations for each darkening process were carried out by one-way analysis of variance (ANOVA), followed by the Duncan’s multiple range test (*p* < 0.05) using Statistica software version 7.0 (Statistica for Windows, Tulsa, OK, USA). In the sensory analysis, differences were considered significant when confidence intervals (*p* < 0.05) did not overlap.

## 3. Results and Discussion

### 3.1. Effect of KOH Reuse on the Oxidation Process

No effect on the duration of the alkaline treatment was found due to the number of reused KOH solutions (Table 1), with the time ranging between 3 h 15 min and 3 h 40 min. The reused lye was enriched with compounds that diffused from the fruit to the alkaline solution, but it did not influence the strength of the alkali.

It must be noted that the concentration of spent lyes was around half of the initial 0.75 M, the content from the first treatment with fresh alkali being significantly higher than the others (*p* < 0.05). Romero et al. [4] reported that when fresh NaOH solution was used during black ripe olive processing, the amount of alkali that penetrated inside the fruits for the same batch of olives was independent of the initial concentration and the temperature at which the process was carried out. However, when the KOH lyes were reused, this did not happen.

It is worth noting that reuse of the new KOH solutions should be a common industrial practice in the future, due to the savings in water and alkali consumption. Taking into account that during these experiments and in current industrial processing the consumption of water in black ripe olive processing is around 4 L/kg olive (0.5 L preservation liquid, 1 L lye, 0.5 L first washing, 1 L second washing and 1 L color fixation solution), the estimated reduction in water consumption would be around 20% after four reuse cycles of the lye, with a similar reduction in the wastewater generated.

In this process, the KOH used was the solution implemented initially plus the quantity added in each new oxidation cycle (Table 1); this means that 26.8 g of KOH/kg of olives were consumed, whereas if the reuses were not carried out, 42 g of KOH/kg of olives would be needed. In other words, there was a saving of 36.2% in this coadjutant, avoiding its discharge.

By contrast, the reuse of the KOH solutions gave rise to a lower weight increase in the fruit than when using fresh alkali, which was also observed during the reuse of NaOH solutions [19].

### 3.2. Physico–Chemical and Sensory Characteristics of Olives

The surface color of the olives after two months of packaging was statistically the same, regardless of having used fresh lye or any of the reused solutions (Table 2).

However, there were differences in the firmness of the olives, where those treated with fresh lye were statistically softer (*p* < 0.05) than the fruits processed with reused KOH solution (Table 2). This phenomenon was also observed during processing with reused NaOH solutions [5] and it may be due to the fact that the degradation of the cell wall by the alkali is somewhat inhibited by the substances diffused from the olives to the reused solution. As could be expected, the potassium content in the final product was much higher (about 6–7 times more) than that found in olives processed with NaOH [20], and this amount increased with the number of reused KOH cycles. This may be due to the fact that a higher potassium content was found in the reused solutions than in the fresh (Figure 1), whereas a lower alkali strength was detected (Table 1), meaning that olives were enriched with this mineral due to its interaction with other olive compounds. Although the concentration of potassium in the washing waters was much lower than in the spent lyes, it was also observed that there was a trend towards its content being increased with the number of reused KOH cycles, in particular for the first washing (Figure 2). 

With regard to the sensory evaluation of the final product, the testers did not observe differences in any of the negative, gustatory or kinesthetic sensations between olives treated with fresh lye and any of those treated with reused lyes (Table 2). To confirm this, triangular differentiation tests were performed between the olives treated with fresh lye and with those from lyes reused four times. Table 2 shows that the testers indicated 15 answers as correct judgments, with more than 22 correct answers being needed to confirm that the samples were different (*p* < 0.05) for the 48 triangular tests performed [14].

All black ripe olives from the experiences could be, therefore, considered as “extra” category [12] since the median of “negative sensations” had a value less than 3.0, close to 1.0–1.2 (Table 2).

The low values for the salty and acidic gustatory sensations were considered normal for this type of elaboration as the NaCl concentration of the cover brine at equilibrium was lower than 2.3% (*w*/*v*) and the pH close to 7 units. As expected [21], the bitter sensation had low values, which was logical as the bitter compounds were eliminated during the oxidation process [3]. In addition, no statistical differences in this gustatory sensation were found among the different treatments, although it has been reported that a high concentration of potassium in olives may imply bitterness in the fruit [22].

With regard to kinesthetic sensations, the testers did not find statistical differences (*p* < 0.05) in hardness despite them being observed with the objective measurements made by shear-compression between fruits treated with reused and fresh lye (Table 2). Additionally, no differences were found for the sensations of fibrousness or crunchiness.

It is noteworthy that the values of the gustatory and kinesthetic sensations were similar to those of commercial olives elaborated using NaOH [21].

### 3.3. Characteristics of Wastewaters and Their Concentrates

As can be seen in Figure 2, the concentration of potassium in the reused alkaline solutions, and first and second washings, exceeded 14,000 mg/L, 6000 mg/L and 3000 mg/L, respectively, whereas the content in the color fixation solutions was always lower than 600 mg/L. After this, only the lyes and washing waters were collected for a further vacuum concentration.

Table 3 shows the physico–chemical characteristics of the global discharges from the initial darkening processes (fresh lye and two washing waters) and after four reuses of the lyes along with their concentrates.

The concentration of glucose and fructose sugars were not high in the global wastewater due to their consumption during olive preservation [23], while the content of mannitol remained high [24].

It is interesting to note the high concentration of acetic acid in comparison with that of lactic acid, as the former was present in the preservation liquid that was reused in the first washing and was also leached from olives during the darkening stage. On the other hand, the concentration of phenolic compounds was low because they were oxidized to achieve the black color of the olives [3].

With regard to the minerals, the global wastewaters presented a low content of nitrogen and sodium and a high content of carbon and potassium, which suggests their possible application for agricultural purposes.

The evaporation of global wastewaters under vacuum reduced the initial volume to one tenth, which means that most of the compounds increased their concentration by values close to 10 times that of the initial concentration. This operation produced a reduction in water activity from 0.99 in the global discharge to around 0.95 in the concentrates, which should not be considered as limiting the growth of microorganisms. However, the high content of phenolic compounds and organic acids in the concentrates would probably contribute to the stability of these solutions. Ethanol was completely evaporated (Table 3) and the concentration of acetic acid was also lower in the concentrate than expected, indicating the presence of these two substances in the distillate phase [25]. The pH of the global wastewater was around 10–11 units regardless of the number of times the lyes were reused, and it was corrected in the concentrates up to 5 units in order to comply with the limits established for organo-mineral fertilizers [26]. Obviously, the content of nitrogen of these solutions increased to a large ex-tent, which was also suggested for liquid fertilizers obtained from the composting of alperujo [27]. Finally, their carbon and, particularly, potassium contents contributed to the valorization of these concentrates as good candidates for future organo–mineral fertilizers, the concentration of potassium being higher than 5%, thereby able to sub-stitute the synthetic KNO_3_ as has been shown for tomato cultivation [11].

## 4. Conclusions

The reuse of KOH solutions (lyes) up to four times gave rise to a product with similar physico–chemical and organoleptic characteristics as the product obtained from fresh solutions. In addition, the application of this new process reduced coadjutant consumption by 32% and water by 20%, which may compensate for the higher cost of KOH in comparison with NaOH.

Likewise, the global liquid from the reused lyes and washing waters contained, after vacuum evaporation, a content in potassium higher than 5%, which makes it a good candidate for use as a fertilizer by replacing commercial potassium nitrate.

## Figures and Tables

**Figure 1 foods-11-01749-f001:**
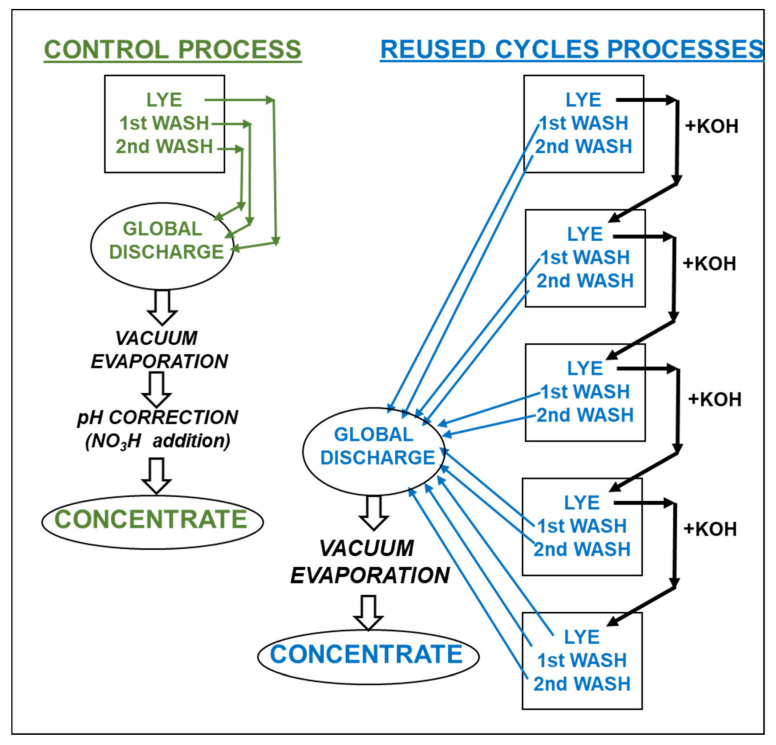
Experimental design.

**Figure 2 foods-11-01749-f002:**
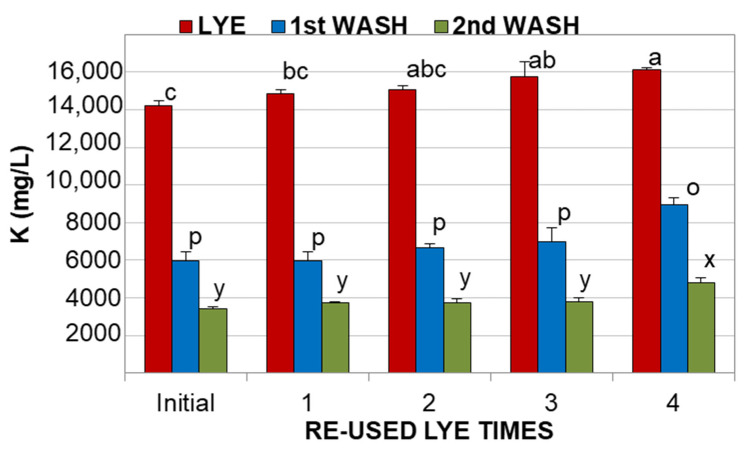
Potassium concentration in lyes and washing waters. Different letters on the bars for each solution (a, b, c for lye; p and o for 1st washing; x and y for 2nd washing) means significant differences according to Duncan’s multiple range test (*p* < 0.05).

**Table 1 foods-11-01749-t001:** Main parameter characteristics of alkaline treatments of olives.

	Reused Cycles				
	Initial	1	2	3	4
Duration of the alkali treatment ^a^	3 h 15 min	3 h 30 min	3 h 40 min	3 h 15 min	3 h 10 min
Initial concentration of KOH solution (M)	0.75	0.75	0.75	0.75	0.75
Final concentration of KOH solution (M)	0.39 (0.01) ^b^ a ^c^	0.36 (0.01) b	0.34 (0.01) bc	0.33 (0.01) c	0.33 (0.01) c
Weight increase in fruits (g/kg)	51.7 (2.4) a	43.0 (3.8) b	41.3 (0.5) b	45.8 (1.2) b	44.7 (0.2) b

Note: ^a^ Time to reach the pit; ^b^ Standard deviation of duplicates in parenthesis; ^c^ values in each row followed by different letters are significantly different according to Duncan’s multiple range test (*p* < 0.05).

**Table 2 foods-11-01749-t002:** Physico–chemical characteristics and sensory evaluation of black ripe olives processed re-using KOH solutions at two months after packaging. Analyses were carried out at two months after packing.

		Reused Cycles				
		Initial	1	2	3	4
PHYSICO–CHEMICAL:						
Superficial color (R_700_)		4.11 (0.03) ^a b^	4.16 (0.11) a	4.18 (0.07) a	4.16 (0,11) a	4.07 (0.04) a
Firmness (N/100 g pitted olive)		1385 (15) b	1515 (23) a	1561 (34) a	1481 (1) a	1518 (29) a
K in flesh (mg/kg)		1430 (36) c	1528 (39) bc	1582(55) b	1648 (27) b	1807 (27) a
SENSORIAL:						
Negative sensations	Abnormal flavor	1.0 (0.0) ^c^ a ^d^	1.0 (0.0) a	1.1 (0.1) a	1.2 (0.1) a	1.1 (0.1) a
Gustatory sensations	Salty	5.9 (0.4) a	5.8 (0.8) a	6.0 (0.3) a	5.5 (0.7) a	6.1 (0.5) a
	Bitter	1.8 (0.5) a	2.1 (0.6) a	1.7 (0.3) a	1.7 (0.3) a	1.8 (0.2) a
	Acid	1.2 (0.2) a	1.3 (0.1) a	1.3 (0.2) a	1.4 (0.2) a	1.4 (0.3) a
Kinesthetic sensations	Hardness	4.7 (0.4) a	4.9 (0.3) a	5.1 (0.4) a	4.9 (0.3) a	4.8 (0.3) a
	Fibrousness	5.0 (0.3) a	5.1 (0.4) a	5.1 (0.2) a	5.3 (0.3) a	4.9 (0.3) a
	Crunchiness	4.1 (0.4) a	4.3 (0.3) a	4.3 (0.2) a	4.1 (0.3) a	4.2 (0.3) a
Correct judgments in triangular test						15 ^e^

Note: ^a^ average and standard deviation of duplicates in parenthesis on physico–chemical parameters; ^b^ values in each row followed by different letters are significantly different according to Duncan’s multiple range test (*p* < 0.05) on physico–chemical parameter; ^c^ Median and robust standard deviation in parenthesis (*n* = 8) on sensorial analysis; ^d^ In each row, median values followed by the same letter do not differ at 5% level of significance because of superimposing confidence intervals; ^e^ number of correct judgments in the 48 triangular comparisons made to detect differences in the flavor between the olives treated with the lye re-used 4 times and with fresh KOH solution (Initial).

**Table 3 foods-11-01749-t003:** Characteristics of the global wastewater (lye + washing waters) and its concentrate up to 10% of the initial. The pH of the concentrates was adjusted to around 5.0 units with HNO_3_.

	Initial Process		After 4 Reuses	
	Global Wastewater	Concentrate	Global Wastewater	Concentrate
Density (kg/L)	1.01	1.12	1.02	1.14
Total solids (g/kg)	36.7	276.9	29.5	249.8
Water activity(aw)	0.992	0.951	0.991	0.941
pH	10.9	5.25	10.5	5.11
Sugars (g/kg):				
Glucose	0.6 (0.0) ^a^	5.6 (0.2)	0.2 (0.0)	1.9 (0.3)
Fructose	0.4 (0.0)	3.9 (0.3)	0.4 (0.0)	3.5 (0.3)
Mannitol	3.6 (0.1)	34.0 (0.8)	6.0 (0.2)	58.9 (1.6)
Acids (g/kg):				
Lactic	0.1 (0.0)	0.9 (0.1)	0.1(0.0)	1.0 (0.2)
Acetic	5.2 (0.2)	27.4 (1.4)	4.8 (0.0)	20.4 (0.6)
Ethanol (g/kg)	1.2 (0.0)	N.D. ^c^	0.5 (0.0)	N.D.
Minerals (g/kg):				
K	6.9 (0.8)	58.4 (0.3)	6.4 (0.5)	55.4 (0.5)
Na	0.5 (0.0)	5.2 (0.3)	0.4 (0.1)	4.4 (0.3)
C	6.6 (0.0)	65.2 (1.0)	6.8 (0.0)	60.1 (1.0)
N	0.2 (0.0)	13.5 (0.2)	0.2 (0.2)	13.5 (0.3)
Phenols (mg/kg):				
Hydroxityrosol	35 (1.0)	340 (18)	28 (1)	301(56)
Tyrosol	75 (2)	704 (45)	107 (12)	993 (59)
Others ^b^	22 (1)	342 (26)	41 (1)	457 (23)

Note: ^a^ Average values and standard deviation in parenthesis; ^b^ Other phenols: sum of glycol, tyrosol glucoside, cafeic and p-cumaric acids; ^c^ not detected.

## Data Availability

No new data were created or analyzed in this study. Data sharing is not applicable to this article.

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
