# Peer review of "Reuse of KOH Solutions during Black Ripe Olive Processing, Effect on the Quality of the Final Product and Valorization of Wastewaters as Possible Fertilizer Product"

_foods, 2022, doi:10.3390/foods11121749_

Round 1
Reviewer 1 Report
The manuscript describes the regeneration and later the proposed reuse of lye and water washing solutions used in the preparation of black olives. The subject is particularly important on the circular economy concept and on reducing wastewater from food processing industries.
The information is valuable, adds to the technical data needed for the industries, and adds value to the food processing industry.
There are just a few suggestions for the manuscript.
In Table 3, there is no indication of the differences between columns two and three and four and five. Are replicates? or different materials?
In figure 2, I suggest using the same letters (a, b, c) for differences in the different cycles, as is usual in mean analysis.
Since the use as fertilizer is a proposed use but not tested in the actual work, I suggest removing it from the title.
Author Response
In figure 2, I suggest using the same letters (a, b, c) for differences in the different cycles, as is usual in mean analysis.
For a better knowledge of the differences, the statistical study was carried out independently for each of the solutions (lye, 1st washing and 2nd washing). That is why those letters have been used. To clarify this, the legend of the figure has been modified:
“Figure 2. Potassium concentration in lyes and washing waters. Different letters on the bars for each solution (a, b, c for lye; p and o for 1st washing; x and y for 2nd washing) means significant differences according to Duncan´s multi range test (p< 0.05).”
Since the use as fertilizer is a proposed use but not tested in the actual work, I suggest removing it from the title.
We have to say that reviewer 2 has indicated that we should include the word product after fertilizer. Hence, in order to combine both proposals, the word "possible" has been included in the title: “Reuse of KOH solutions during black ripe olive processing. Effect on the quality of the final product and valorization of wastewater as possible fertilizer product”Thank you very much for the suggestions that have undoubtedly improved the text

Reviewer 2 Report
Minor revision
Title to be changed to “Reuse of KOH solutions during black ripe olive processing. Effect on the quality of the final product and valorization of wastewater as fertilizer products”
row 4: cover brine [8]– consider revising to: cover brine [8]
row 16: replace “to that obtained” with “to the one obtained”
row 37: Remove “As can be deduced from the foregoing”
row 45: replace “along with the first washing” with “alongside the first washing”
row 48: add “as a packing cover brine”
row 51: replace “agronomic” with “agricultural”
row 61: replace “Olives processing” with “Olive processing”
row 77-78: replace “and after the 77 fixing solution application.” with “and after applying the fixing solution.”
row 90: replace “in another two tanks,” with “in two other tanks”
row 131-135: consider text alignment JUSTIFY
row 136: Detailed HPLC analysis is required.
row 255: replace “agronomic” with “agricultural”
Author Response
Title to be changed to “Reuse of KOH solutions during black ripe olive processing. Effect on the quality of the final product and valorization of wastewater as fertilizer product”
Reviewer 1 has proposed to remove the word fertilizer from the title. Hence, in order to combine both proposals from reviewers 1 and 2, the word "possible" has been included in the title: “Reuse of KOH solutions during black ripe olive processing. Effect on the quality of the final product and valorization of wastewater as possible fertilizer product”
row 4: cover brine [8]– consider revising to: cover brine [8]
This sentence is not in this row 4 but in number 48 and it has been modified
row 16: replace “to that obtained” with “to the one obtained”
Reviewer's suggestion accepted
row 37: Remove “As can be deduced from the foregoing”
Reviewer's suggestion accepted
row 45: replace “along with the first washing” with “alongside the first washing”
Reviewer's suggestion accepted
row 48: add “as a packing cover brine”
Reviewer's suggestion accepted
row 51: replace “agronomic” with “agricultural”
Reviewer's suggestion accepted
row 61: replace “Olives processing” with “Olive processing”
Reviewer's suggestion accepted
row 77-78: replace “and after the 77 fixing solution application.” with “and after applying the fixing solution.”
Reviewer's suggestion accepted
row 90: replace “in another two tanks,” with “in two other tanks”
Reviewer's suggestion accepted
row 131-135: consider text alignment JUSTIFY
Reviewer's suggestion accepted
row 136: Detailed HPLC analysis is required.
Detailing the 3 methods of analysis would imply greatly expanding the text. For this reason, a summary with the most relevant characteristics of the methods has been included:
“Sugars were analyzed by HPLC with A Rezex RCM-Monosaccharide Ca+ (8%) column (300 × 7.8 mm i.d., Phenomenex) held at 85 °C, deionized water as eluent at 0.6 ml/min and the detection was performed with a Waters 410 refractive index detector [16]. Organic acids and ethanol were analysed using the same detector with a Spherisorb ODS-2 (5 μm, 250 × 4.6 mm, Waters Inc.) column with deionised water (pH adjusted to 2.3 with phosphoric acid) as mobile phase [17]. The same column was used to separate the phenols that are identified by a Waters 996 diode array detector (Waters Inc., Mildford, MA) using an elution gradient with water (adjusted to pH 3.0 with phosphoric acid) and methanol [18].”
row 255: replace “agronomic” with “agricultural”
Reviewer's suggestion accepted
Thank you very much for the suggestions that have undoubtedly improved the text
Reviewer 3 Report
In this manuscript, authors describe their study on the reuse of KOH solutions used for black ripe olive processing.
The argument presented is very interesting and deals with an important aspect relating to innovative processes for reducing environmental impact.
The presentation of the results is clear both in the text and in the tables.
Just little comment below.
Affiliation: it is always the same, so put 1 for all authors
Section 2.4: how the sample is treated after digestion and recovered before the instrumental reading?
Section 2.5: please specify for each analysis the code of the method APHA SMEWW adopted.
Line 234: Decide whether or not to use the thousands separator

Author Response
Affiliation: it is always the same, so put 1 for all authors
The reviewer's suggestion cannot be taken into account because the numbers are later related to the authors' emails.
Section 2.4: how the sample is treated after digestion and recovered before the instrumental reading?
The sentence has been expanded to include more details of the analysis:
One mL of liquid or 1 g of olive paste was digested in a DigiPREP equipment (Quebec, Canada) with 25 mL of 14 M HNO3 at 120ºC for 8 hours; nitric was evaporated at 140 ºC after addition 5 mL solution HClO4/HNO3 (4/1). Subsequently, solution was put into a 25 mL graduated flask and filled with deionized water and diluted to obtain a concentration lower than 200 meq/L of sodium or potassium in order to be determined in a Metheor flame photometry (model NAK-1, PACISA, Spain).
Section 2.5: please specify for each analysis the code of the method APHA SMEWW adopted.
Method code has been included as suggested by the reviewer
“Total solids were determined according to 2540 method of standard procedures [15],”
Line 234: Decide whether or not to use the thousands separator
Removed thousands separator
Thank you very much for the suggestions that have undoubtedly improved the text